# The Influence of Affective Empathy on Online News Belief: The Moderated Mediation of State Empathy and News Type

**DOI:** 10.3390/bs14040278

**Published:** 2024-03-27

**Authors:** Yifan Yu, Shizhen Yan, Qihan Zhang, Zhenzhen Xu, Guangfang Zhou, Hua Jin

**Affiliations:** 1Department of Psychology, Tianjin Normal University, Tianjin 300387, China; yuyifan@skiff.com (Y.Y.);; 2School of Health, Fujian Medical University, Fuzhou 350122, China; yanshizhen@fjmu.edu.cn

**Keywords:** affective empathy, online news, fake news, news belief

## Abstract

The belief in online news has become a topical issue. Previous studies demonstrated the role emotion plays in fake news vulnerability. However, few studies have explored the effect of empathy on online news belief. This study investigated the relationship between trait empathy, state empathy, belief in online news, and the potential moderating effect of news type. One hundred and forty undergraduates evaluated 50 online news pieces (25 real, 25 fake) regarding their belief, state empathy, valence, arousal, and familiarity. Trait empathy data were collected using the Chinese version of the Interpersonal Reactivity Index. State empathy was positively correlated with affective empathy in trait empathy and believability, and affective empathy was positively correlated with believability. The influence of affective empathy on news belief was partially mediated by state empathy and regulated by news type (fake, real). We discuss the influence of empathy on online news belief and its internal processes. This study shares some unique insights for researchers, practitioners, social media users, and social media platform providers.

## 1. Introduction

Online news is characterized by its quickness, multimedia, and interaction. Online news quickly informs recipients of news events happening worldwide and promotes more news content or information conforming to existing attitudes and hidden prejudices. In recent years, the Internet and social media have developed rapidly. By December 2022, the number of netizens in China reached 1.067 billion, and the Internet penetration rate reached 75.6%. With the rapid development of the Internet and social media, application software such as TikTok gradually changed from entertainment to information platforms, becoming essential channels for netizens to obtain news. Therefore, online news belief has become an important social issue and academic focus.

In research fields on online news belief, a common method employed is the News Evaluation Task. This method randomly presents participants with a set of real and fake news headlines. Participants are required to read each news item and assess its believability or authenticity by assigning Likert scale scores [1,2,3,4,5].

News belief involves complex cognitive processing; it has been mainly explored from two aspects: news characteristics and recipients’ individual differences. Regarding news characteristics, the higher the credibility of the source [6,7] and the greater one’s familiarity [3,8] with the news, the more accurately participants perceived its believability. Concerning individual differences, previous research found that differences in personal traits may affect news belief, such as personality [8,9,10,11,12], thinking style [4,13,14,15], media literacy [16,17,18], and prior attitudes [19,20,21,22].

The emotional contagion theory posits that emotional sharing between individuals forms the basis of empathy [23]. Individuals with strong empathic abilities have strong emotional sharing abilities [24]. Therefore, we speculated that individuals with high trait empathy have a stronger ability to share emotions, causing them to develop a more immersive understanding and resonance with news content and belief in online news content. In other words, an individual’s empathy may influence their belief in online news. However, less attention has been devoted to the impact of empathy on online news belief and its internal processes to date. This study focused on the effects of experiencing specific emotions on belief in fake news, i.e., the role of empathy in online news belief. We aimed to address the following questions: Firstly, do individual differences in trait empathy influence news belief? If so, what are the internal processes behind this influence?

### 1.1. Trait Empathy and the Online News Belief

Trait empathy (TE) refers to “the ability to share the feelings and experiences of others by imagining their situation” [25]. It is a stable personality trait with significant social functions, making it a focal point of research in psychology. The two primary components are cognitive empathy (CE) and affective empathy (AE). CE refers to the ability to engage in cognitive role-taking or the cognitive processes involved in adopting others’ psychological perspectives. AE involves responding emotionally to others’ experiences or sharing the feelings of a companion [26].

Limited research suggests that TE may influence online news belief. For example, Martel et al. [1] found that a higher level of emotional arousal before reading news predicted greater belief in fake (rather than real) news. Although this study did not directly measure individual TE, its emotional indicators were somewhat related to individual TE when the Positive and Negative Affect Schedule (PANAS) scale was used to assess participants’ emotional state before reading news. Preston et al. [27] found that participants with higher emotional intelligence (attending to emotions, emotion-based decision-making, and empathic concern) were less likely to be misled by fake news. Their fake news detection task combined objective judgment, professionalism, argument strength, and belief in news items. This novel and comprehensive fake news detection task unintentionally engages participants’ analytical thinking, which could directly influence news believability [4,9,13,20]. In many real-world situations, recipients are unlikely to engage in this kind of analytical thinking when encountering fake news. Furthermore, they did not explicitly separate the roles of two different empathy components in news belief. Therefore, more experimental evidence is needed to elucidate the relationship between empathy and online news belief.

This study directly investigated the impact of different TE components on online news belief, using participants’ judgment of news believability as the primary measure. Given that affective empathy rather than cognitive empathy is more relevant to people’s behaviors (such as prosocial behavior and moral decision making) [28,29,30], we propose Hypothesis 1: an individual’s AE, rather than CE, would be more related to their belief in online news. Specifically, higher scores in AE would be associated with higher belief in online news.

### 1.2. The Mediating Effect of State Empathy on Affective Empathy—The Believability of Online News

State empathy (SE) refers to the emotional response that individuals experience when imagining or observing the emotional state or circumstances of others [31]. Empathy in daily life often occurs within specific contexts and is influenced by situational factors. Rusting [32,33] suggested that the influence of TE on emotion processing is mediated by SE; in other words, SE allows TE to be expressed. The impact of TE on individual behavior is most evident when expressed in certain contexts. SE in specific situations or tasks may reflect either stimulus or task-induced state effects or more stable TE. 

Similar to many emotion-related processes, some empathy components occur implicitly and sometimes without awareness. State empathy in news reading involves such emotional processing. News reading may cause the phenomenon of emotional contagion, the tendency to automatically mimic and synchronize facial expressions, vocalizations, postures, and movements with those of another person and, consequently, emotional convergence with the other [34]. An extensive body of literature on emotion explains the differential impact of negative and positive mood states and specific discrete emotions on cognition and decision making. The literature on the relationship between emotion and belief found that emotions induced by news reading can influence belief [19,35]. For instance, a negative mood state generally increases skepticism, whereas a positive mood state increases gullibility and decreases the ability to detect deception [36,37]. Martel et al. [1] found that heightened emotionality in participants predicted greater belief in fake (but not real) news posts. Bago et al. [35] asked participants to read 16 news headlines categorized into four groups: fake news consistent with the Republican party, real news consistent with the Republican party, fake news consistent with the Democratic party, and real news consistent with the Democratic party. They then required participants to assess the authenticity of these headlines and report their emotional states while reading. The results indicated that emotional experiences during reading (excluding anger) diminished political news credibility. Wang et al. [38] uncovered a positive correlation between belief in misinformation about food safety and negative emotions. Negative emotions partially mediated the relationship between misinformation and its subsequent diffusion on social media and completely mediated the relationship between misinformation and face-to-face diffusion among high-trust individuals. 

Therefore, when investigating the relationship between TE and online news belief, SE was considered essential. Trait empathy may be expressed through state empathy, which directly influences online news belief. We propose Hypothesis 2: SE (characterized by an emotional response following headline reading) is related to online news believability. SE is expected to partially mediate AE’s effects on online news belief. Specifically, higher scores in AE would be associated with higher SE in reading online news, and higher scores in SE would increase the believability of online news.

### 1.3. The Moderating Effect of News Type on Affective Empathy-State Empathy—Online News Belief 

Piksa et al. [39] found that those who believed in real news also believed in fake news, suggesting no relationship between belief and news type. Most of their results indirectly suggest that belief may be influenced by the type of news. Studies focusing on individual differences have found that cognitive style only affects the accuracy of perceived fake news but not the accuracy of perceived real news [2,4,15,20]. For instance, actively open-minded individuals and those showing more reflective and less intuitive thinking patterns are more likely to consider fake news less reliable. However, these traits do not correlate with reaction scores to real news [20]. Four groups of participants with varying susceptibilities to (fake) information showed significant differences in their accuracy in perceiving real news from fake news [39]. Moreover, studies examining factors influencing online news believability [1,3,40,41] or intervention methods [5,42,43] have observed similar effects.

Furthermore, Bago et al. [35] found no significant differences in emotional responses among participants reading real and fake news. However, emotional experiences while reading weakened belief in fake news more than real news. Thus, news type was expected to moderate the relationship between SE and online news belief. Martel et al. [1] found that higher emotional arousal before news exposure predicted greater trust in fake (but not real) news. Therefore, we propose Hypothesis 3: the influence of SE on online news belief is moderated by news type. Specifically, higher scores in SE would increase the believability of both fake news and real news, but the effect on fake news is even greater.

In conclusion, using a fake news detection task similar to that used by Pennycook et al. [3], we aimed to investigate the impact of TE on news belief and its internal processes. The participants’ TE scores, comprising CE and AE dimensions, were assessed using the IRI-C scale. Participants were instructed to read real and fake news and evaluate measures such as emotional response and belief in each news item. News headlines were presented in a format similar to online news, complete with headlines, images, and accompanying descriptive text. To ensure a content-independent understanding of the relationship between empathy and online news belief, the topics encompassed a wide range, including social issues, current events, and scientific knowledge.

## 2. Materials and Methods

### 2.1. Participants

A total of 149 undergraduates were recruited. The study included 140 participants (47 males, 19.67 ± 2.16 years) after excluding 9 participants who did not complete all experiments diligently. All participants had normal or corrected-to-normal vision and normal color vision. All participants signed an informed consent form and received payment for their participation. The present study was approved by the Ethics Committee of Tianjin Normal University (No. 2023091104).

### 2.2. Materials

There were 100 news headlines. Considering the interference of the experimental duration on the data quality, the headlines were divided into two sets that matched valence (3.23 ± 1.24 vs. 3.01 ± 1.03, *t*(49) = 1.103, *p* = 0.275), arousal (4.09 ± 0.48 vs. 4.07 ± 0.50, *t*(49) = 0.184, *p* = 0.855), familiarity (3.22 ± 0.39 vs. 3.22 ± 0.41, *t*(49) = −0.109, *p* = 0.914), empathy (4.66 ± 0.54 vs. 4.75 ± 0.56, *t*(49) = −1.158, *p* = 0.252), and believability (5.17 ± 0.85 vs. 5.05 ± 0.68, *t*(49) = 1.207, *p* = 0.253). Each set included 25 real and 25 fake headlines. All fake news headlines were initially sourced from a well-known Chinese fact-checking website (https://www.piyao.org.cn/pysjk/frontsql.html (accessed on 1 January 2020.)). Real news headlines were selected from mainstream news sources (e.g., www.cctv.com (accessed on 1 January 2020.)). Materials were presented in the format of typical online news articles (including headlines, images, and descriptive text). The material topics covered social issues, current events, and general scientific knowledge. All news images were standardized to a size of 1250 × 780 pixels using Adobe Photoshop 2021 (Figure 1 shows examples of news stimuli). More details can be found in the Appendix A.

The IRI-C [44] comprises 22 items and is rated on a 5-point Likert scale, from 0 (does not describe me well) to 4 (describes me very well). Higher scores indicate a higher empathy level. The scale is divided into four subscales: Perspective Taking (PT), Fantasy Scale (FS), Empathetic Concern (EC), and Personal Distress (PD). PT and FS fall under CE, while EC and PD are related to AE. Cronbach’s α for the scale was 0.753. The example items for the main IRI-C scale were as follows. PT: I try to see each person’s side of the argument before making a decision; FS: After watching a play or movie, I feel as if I were one of the characters in the play; EC: Other people’s misfortunes do not usually bother me much; PD: When I see that someone has had an accident and needs help badly, I become so nervous that I almost have a nervous breakdown.

### 2.3. Procedure

The experiments were programmed using E-prime 3.0 software. The news was presented in the center of the computer screen in pseudo-random order. The number of real and false news headlines displayed was balanced among participants. The participants were instructed to read the content of each news headline thoroughly and then report their feelings by pressing the number keys on the keyboard, which included their emotional response (1—Cannot feel it at all; 7—Can totally feel it), believability (1—Extremely unbelievable; 7—Extremely believable), valence (1—Very negative; 4—Neutral; 7—Very positive), arousal (1—Not at all; 7—Extremely), familiarity (1—Not at all; 7—Extremely). The experiment lasted approximately 60 min with one break, after which IRI-C questionnaire data and demographic data were collected.

### 2.4. Data Analysis

(1) Descriptive statistics were employed for variables such as trait empathy, affective empathy, cognitive empathy, state empathy, and believability. Pearson’s correlation coefficient was used to calculate the relationships between these variables.

(2) The Influence of Affective Empathy on News Believability: The Mediating Effect of State Empathy.

A mediation model was constructed to examine AE’s mediating effect using data from all participants. The mediating effect was examined using Hayes’ Bootstrap method [45] and Hayes Process Macro 4.0 in SPSS 29 with a sample size of 5000 and a 95% confidence interval. In the model, AE served as the independent variable (X), belief in the news (believability) as the dependent variable (Y), valence as the control variable (S), and SE as the mediating variable (M).

(3) The moderating effects of news type on state empathy’s mediating role in affective empathy and believability.

Analyses were conducted using the statistical analysis software SPSS 29.0.1 (IBM). A moderated mediation analysis was tested using Hayes Process Macro 4.0 in SPSS 29 with a sample size of 5000 and a 95% confidence interval. Affective empathy served as the independent variable (X), perceived believability (believability) as the dependent variable (Y), state empathy as the mediating variable (M), valence as the control variable (S), and news type (fake news = 0, real news = 1) as the moderating variable (V). Additionally, simple slope tests were conducted.

## 3. Results

### 3.1. Descriptive Results and Correlation Analysis 

Paired samples *t*-tests examined differences in valence, arousal, and familiarity between real and fake news, revealing that fake news had significantly lower valence than real news (2.80 ± 0.69 vs. 3.45 ± 1.39, *t*(49) = −4.12, *p* < 0.001), fake news was similar to real news in arousal (4.04 ± 0.50 vs. 4.12 ± 0.47, *t*(49) = −1.06, *p* = 0.297), and familiarity (3.23 ± 0.50 vs. 3.21 ± 0.28, t(49) = 0.31, *p* = 0.758).

Pearson’s correlation analysis revealed that SE was positively correlated with TE, AE, and believability. TE was positively correlated with its AE and CE components. Believability was positively correlated with TE and AE. AE was positively correlated with CE. The other pairwise correlations were not significant. These results are presented in Table 1.

### 3.2. The Influence of Affective Empathy on News Beliefs: The Mediating Effect of State Empathy

As shown in Figure 2, the results support the hypothesis that SE partially mediates AE’s influence on news believability. The mediation test’s indirect effect did not include zero (Effect = 0.008, SE = 0.004, 95%CI = [0.002, 0.016]). Furthermore, after controlling the mediator variable state empathy, the direct effect was significant, with the confidence interval not including zero (Effect = 0.022, SE = 0.009, 95%CI = [0.004, 0.040]).

### 3.3. The Moderating Effects of News Types on the Mediating Role of State Empathy toward Affective Empathy and Believability

Statistical analysis showed that AE’s influence on believability and the mediating role of news type in the path of AE affecting SE was not valid (*Index* = −0.002, *SE* = 0.007, 95%CI = [−0.015, 0.011]). However, news type had a mediating effect on SE’s path, affecting belief in AE’s influence on believability. The model index does not include zero (*Index* = −0.009, *SE* = 0.005, 95%CI = [−0.019, −0.001]), thus indicating a valid and moderate mediation effect, as shown in Figure 3.

Table 2 presents the mediation effects and bootstrap confidence intervals for different moderating variable (news type) levels in the relationship between AE and believability. The results indicate that SE partially mediates AE’s influence on believability for both real and fake news. 

To further analyze the interaction between news type and state empathy, simple slope tests were conducted to calculate separate effect values for ‘real’ and ‘fake’ news types. The results are illustrated in Figure 4. Higher state empathy toward a news article was associated with higher believability. The SE predicted believability under ‘real’ and ‘fake’ news conditions. However, SE’s predictive effect on believability was significantly greater in the ‘fake’ news condition (*b* = 0.431) than in the ‘real’ news condition (*b* = 0.202).

## 4. Discussion

This study investigated the relationship between TE and belief in online news as well as the mediating and moderating roles of state empathy and news type in that relationship. Our results revealed that state empathy mediated the relationship between AE and belief in online news, and news type moderated that relationship.

### 4.1. The Influence of Trait Empathy on the Believability of News

TE’s influence on cognition has been extensively studied [46,47,48,49,50]. For instance, using a dot–probe paradigm and eye-tracking technique to investigate attention bias toward happy faces and positive words in participants with high and low empathy, Liu et al. [48] found that trait empathy influenced the processing of emotional information. However, few studies addressed questions about TE’s impact on belief in online news. This study found that AE rather than CE within TE positively correlated with belief in online news, supporting Hypothesis 1.

The two different components of TE, CE and AE, reflect different aspects of an individual’s empathetic abilities and have different neural bases [51,52,53,54,55]. It is reasonable that they affect belief in online news differently. Previous studies found that people’s behavior (e.g., altruistic behavior and moral decision-making) is more influenced by AE than CE [28,29]. For example, Herne et al. [29] found that, within TE subscales, only the AE subscale was positively associated with dictator game-giving. The predictive effects of age and gender on altruistic moral decision making were influenced by AE rather than CE [30]. Our findings separated CE and AE roles in online news belief. The internal processes through which AE influences belief in online news may be related to cognitive style. We found that rational thinking is related to news belief [4,13,20]. The dual-process model of empathy posits that CE is a more rational process, whereas AE is less rational [56,57]. For instance, Martingano et al. [58] measured empathy using the Interpersonal Reactivity Index (IRI) and rational thinking tendency using the Need for Cognition Scale (NFC). They also assessed rational thinking using the Cognitive Reflection Task (CRT). Their results revealed a complex relationship between empathy and rational thinking, depending on how rationality (rational thinking tendency or rational thinking performance) and empathy (affective empathy or cognitive empathy) were measured. Nevertheless, rational thinking and affective empathy exhibited a significant negative correlation, while the correlation between rational thinking and cognitive empathy was insignificant.

It seems to be inconsistent with the results of Preston et al. [27]. Preston et al. [27] found that people with higher emotional intelligence, who were more emotionally perceptive, were less likely to fall for fake news. “This finding supports the idea that high-EQ individuals are more likely to be able to see beyond the emotionally charged content, which is a hallmark of fake news, leading to a more effective critical evaluation of the likely veracity of the content.” The conclusion is that higher-order thinking and emotional detection lead to less belief in fake news. This inconsistency may be related to the differences in the specific methods of the two studies. Our focus is trait empathy, which is not the same as emotional intelligence. Additionally, as mentioned in the Introduction, their results may be mixed with the effects of analytical thinking. Hoffman [59] views empathy as a largely involuntary vicarious response to affective cues from another person or her situation; our finding that the irrational component of trait empathy is related to online news belief rather than its rational component also suggests the possible influence of analytical thinking. Therefore, the results of the two studies may not be completely contradictory.

In conclusion, individuals with strong AE may prefer less rational thought [58,60,61]. Thus, high AE individuals are more likely to believe fake news.

### 4.2. The Mediating Effect of State Empathy

We found that empathy partially mediated the relationship between AE and news belief, thus confirming Hypothesis 2. Specifically, higher AE individuals exhibited higher SE levels in the task context, leading to higher news belief.

There is limited research on the relationship between SE and online news belief. Generally, this result was consistent with a few existing studies, particularly those using political news as their material [19,35]. Rijo and Waldzus [19] presented participants with five real and five fake political news headlines extracted from Facebook and asked them to judge their credibility. The participants also had to answer questions related to their political beliefs. Their results showed that belief in the news was influenced by participants’ emotional reactions to different political beliefs during headline reading. Negative beliefs about the political system increased emotional reactions to both real and fake news, subsequently enhancing news believability, accuracy categorization, and willingness to share news. Their findings highlighted the relationship between individualized political beliefs, specific emotional responses, and belief in political news. The present study revealed a relationship between individual differences in TE, SE, and belief in non-political news, whose conclusion may be more general.

### 4.3. The Moderating Effect of News Type

Furthermore, the present study revealed that news type moderated SE’s effects on belief. Specifically, SE significantly predicted the belief of both real and fake news, significantly affecting belief in fake news. It is suggested that higher AE individuals are more likely to believe the news, especially fake news, partially because of their emotional responsiveness when reading the news. 

Although previous research did not explicitly investigate the moderating role of news type in online news belief, many studies on fake news have found that certain factors impact belief in different news types [5,41], as mentioned in the Introduction. For instance, Calvillo et al. [5] investigated the impact of truthiness cues on news believability and found that they effectively reduce the repetition effect in fake news believability, with news type playing a significant moderating role. Martel et al. [1] found that specific emotions (such as interest, excitement, fear, tension, etc.) expressed before reading news headlines significantly predicted higher belief in fake (but not real) news. Additionally, compared to a control induction or a reason induction, emotion induction led to higher belief in fake but not real news. From the perspective of empathy, our study clarified the role of news type in news belief, elucidating how news type moderated the AE–SE–news belief relationship.

Additionally, our study found that SE predicted belief in fake news to a greater extent than real news. This finding indicated that empathy substantially influences fake news’ believability, with highly empathetic individuals more likely to trust fake news. In real-life contexts, fake news often carries a higher emotional charge. Emotional economics theory asserts that fake news creators intentionally write stories that evoke emotions on social media platforms to attract attention and generate revenue [62]. Most emotions in news articles show statistically significant differences between real and fake news [63]. Therefore, individuals with high affective empathy should be cautious when using social media, particularly with emotionally charged news, and evaluate news content’s credibility consciously. In everyday life, empathetic individuals, when seeing news about an event that is described in a lot of emotional words, would better have the following conversation with themselves: The author seems to be describing the incident with emotion, and the truth of the incident may not be exactly like this? Is there something wrong with the content of this news? Additionally, is it possible that this is happening in reality? Such self-questioning may trigger one’s analytical thinking and avoid the increased belief in fake news caused by empathy.

### 4.4. Limitations and Future Research

However, the present study has some limitations. Firstly, caution is recommended when extrapolating our results. Our participants were native Chinese, and there were fewer male participants than females. Cultural and gender differences can influence emotional responses and belief in online news. Secondly, there is a lack of direct evidence for our approach to explaining AE’s effect on belief. Future studies should include a Cognitive Reflection Test [64] to investigate the relationship between empathy, analytical thinking, and belief in online news. Finally, the reliability of IRI-C, although sufficient, was relatively low.

Additionally, future research can expand on current findings in several ways: (1) neuroscientific investigations: Examining the neural basis of the relationship between empathy and online news believability with functional magnetic resonance imaging (fMRI). This method can provide objective evidence at the neural level of how affective empathy, rather than cognitive empathy, influences online news believability. (2) As this study highlighted empathy’s crucial role in online news believability, future research could explore related variables such as an individual’s self-construal orientation or the news situation context (e.g., whether the subject belongs to an in-group or an out-group). Exploring these factors may reveal new insights. (3) We would examine factors that influence online news believability using analogies to research initial trust. Research in this area could explore cognitive resources, intergroup interaction contexts, and more macro-level factors. This broader perspective may uncover new dimensions of influence. (4) In this study, valence was included as a covariate but not examined as an independent variable. Future research could independently manipulate valence to explore its direct impact on news belief, providing supplementary evidence. These suggested avenues of research can further enrich our understanding of how empathy and related factors influence online news believability in the digital age.

## 5. Conclusions

Affective empathy, rather than cognitive empathy, influenced online news believability.Concerning affective empathy’s impact on news believability, state empathy acted as a partial mediator.News type moderated state empathy’s effects on belief. Furthermore, state empathy predicted belief in fake news to a greater extent than belief in real news. Our findings shed light on empathy’s influence on online news believability and its internal processes. They also provide a possible strategy to reduce belief in fake news. We found that individuals with higher levels of affective empathy may be more susceptible to fake news, particularly news that elicits high levels of state empathy. Our study underscores the importance of discerning authenticity in news, especially when exposed to news that triggers a high state of empathy.

## Figures and Tables

**Figure 1 behavsci-14-00278-f001:**
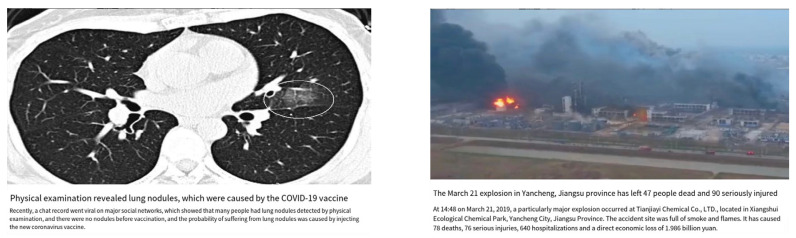
The news stimuli (Left: fake; Right: real).

**Figure 2 behavsci-14-00278-f002:**
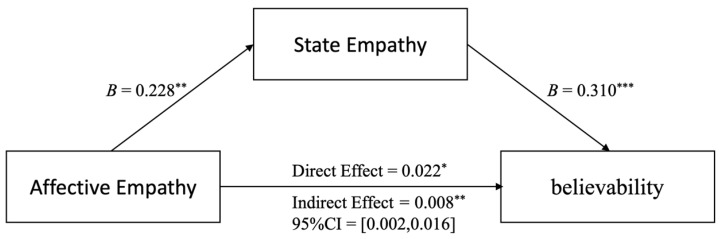
The mediating effect of state empathy is shown. Note: * *p* < 0.05; ** *p* < 0.01; *** *p* < 0.001. The results demonstrated that state empathy partially mediated the influence of affective empathy on news believability.

**Figure 3 behavsci-14-00278-f003:**
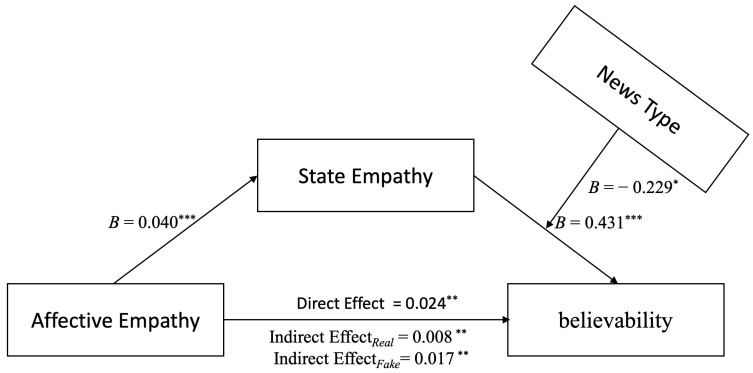
The moderated mediation model. Note: When controlling for the mediating variable, the direct effect of state empathy (empathy level) was statistically significant, with a confidence interval that did not include zero (Effect = 0.024, *SE* = 0.007, 95%CI = [0.011, 0.038]). News type significantly moderated the impact of state empathy on believability (*b* = −0.229, *SE* = 0.095, *p* = 0.017, 95%CI = [−0.416, −0.042]). ** p* < 0.05; *** p* < 0.01; **** p* < 0.001.

**Figure 4 behavsci-14-00278-f004:**
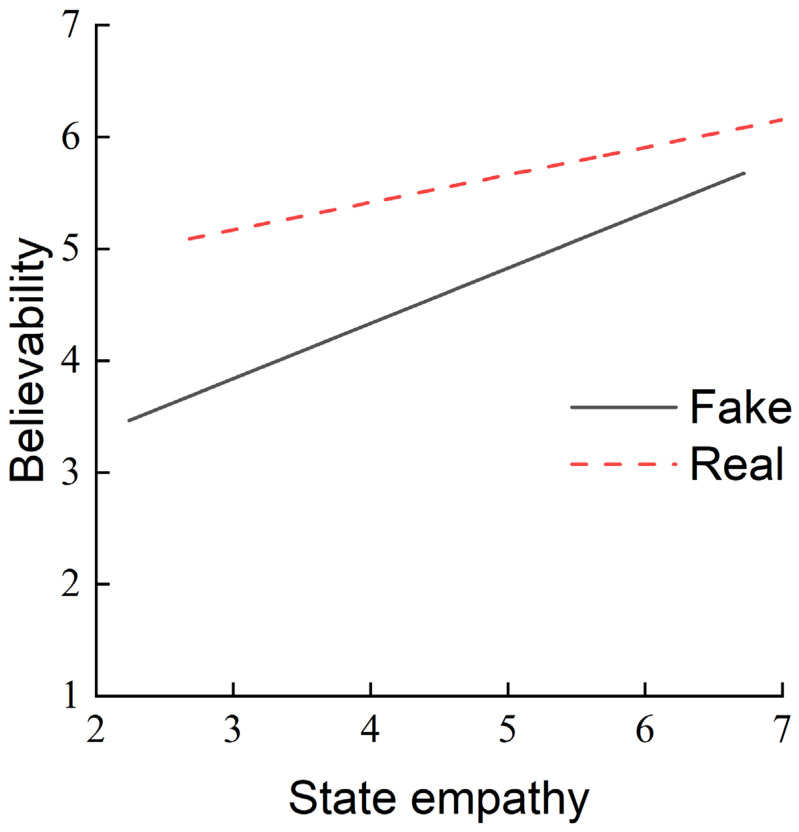
Moderating effect of real and fake types between state empathy and believability. Note: In the “fake” news condition, *b* = 0.431, *SE* = 0.069, *t* = 6.246, *p* < 0.001, 95%CI = [0.295, 0.566]. In the “real” news condition, *b* = 0.202, *SE* = 0.068, *t* = 2.971, *p* = 0.003, 95%CI = [0.068, 0.336].

**Table 1 behavsci-14-00278-t001:** Correlation analysis results between variables.

	TE	AE	CE	State Empathy	Believability
TE	1				
AE	0.718 ***	1			
CE	0.574 ***	0.239 **	1		
State Empathy	0.194 *	0.295 ***	0.064	1	
Believability	0.329 ***	0.337 ***	0.054	0.380 ***	1
M (SD)	56.11 (9.56)	27.10 (6.11)	29.16 (6.25)	4.70 (0.80)	5.11 (0.69)

Note: * *p* < 0.05; ** *p* < 0.01; *** *p* < 0.001.

**Table 2 behavsci-14-00278-t002:** The mediating effect of state empathy between AE and believability in real and fake news.

Moderator		Effect	Boot SE	95%CI
News type	Real	0.008	0.003	[0.002, 0.015]
	Fake	0.017	0.005	[0.009, 0.028]

## Data Availability

The data and analysis code for this study are available at: https://osf.io/nh6uf/?view_only=75d80397fe3f40eebdac1d98668ed17b (accessed on 27 January 2024).

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
