# Peer review of "The Influence of Affective Empathy on Online News Belief: The Moderated Mediation of State Empathy and News Type"

_behavsci, 2024, doi:10.3390/bs14040278_

Round 1
Reviewer 1 Report
Comments and Suggestions for Authors
Thank you for the opportunity to review your manuscript examining the role of empathy in susceptibility to believing fake news. As interest in the research area of emotion and online news processing continues to grow, especially in regards to acceptance of misinformation, I believe there is value to your study. However, the writing and framing of the rationale for your study and the evidence used in the manuscript both need to be strengthened. Simply relying on the argument that empathy has not widely been studied in the context of believing fake news is not enough. Furthermore, there is confusion as to whether the goal of your study is to explain the internal process of empathy or to examine the impact of empathy in believing false news. Your manuscript focuses disproportionality on modeling an internal process of empathy that is already known. If indeed, you are interested in the connection between emotion and news belief, you need to more fully address the areas of literature given below in the introduction of your manuscript and test and report differences in real and fake news believability throughout your results.
Introduction/Literature Review:
Emotion and News Research: In the area of emotion and news, researchers have been interested in 1) how the emotional framing of news content influences readers. For example, research confirms that audiences are more likely to believe emotional news, even if it is false. 2) The effects of experiencing specific emotions on belief in fake news. Your study falls into this latter category. You need to make this clear. At the very end of your manuscript, you say that this about “being exposed to high-state empathetic news” (line 384). You blur the emotion of the news consumer and the emotion of the news here.
Emotion and Belief in Fake News: First, an extensive body of emotion literature explains the differential impact of negative and positive mood states and specific discrete emotions on cognition and decision-making. For example, literature on the relationship between emotion and gullibility has found that a negative mood state generally increases skepticism, whereas a positive mood state increases gullibility and decreases the ability to detect deception (Forgas and East 2008; Forgas 2019). Then, make it clear that specifically in the area of fake news belief, several studies have suggested that people who engage in more reasoning are less likely to fall for fake news. Martel et al. (2020) found heightened emotionality in participants was predictive of greater belief in fake (but not real) news posts. Preston et al. (2021) found people with higher emotional intelligence, who were more emotionally perceptive, were less likely to fall for fake news. “This finding supports the idea that high-EQ individuals are more likely to be able to see beyond the emotionally charged content which is a hallmark of fake news, leading to a more effective critical evaluation of the likely veracity of the content.” [https://www.ncbi.nlm.nih.gov/pmc/articles/PMC7951906/]. The conclusion here is that higher order thinking and emotional detection lead to less belief in fake news.
The Role of Empathy in News Processing: You need to better explain the automatic and more cognitive aspects (perspective taking) of empathy [see: https://pubmed.ncbi.nlm.nih.gov/15537986/] and how the appraisal pattern of this positive, prosocial, “every day mind reading” emotion led to your expected predictions and results. It seems there is reason to believe that the approach, positive mindset of empathy will lead to more susceptibility to fake news, especially for the more automatic affective empathy process and in-the-moment state empathy, as you suggest. Currently, your use of emotional contagion theory is not sufficient. You need to go deeper into the specific emotion of empathy. This is likely not just about being more “emotionally responsive” as you state on line 324.
Hypotheses: Make is clear whether you predict the influence of empathy will relate to an increase or decrease in susceptibility to fake news specifically.
Method:
Stimulus Materials: Can you provide an example of the news stimuli used? You also need to explain if the stimuli were selected to be emotionally neutral or generate emotional responses in participants and the implications of this. You state that you measured emotion response but it is not included in your descriptive results.
Measures: Provide examples items for the main IRC-C trait empathy scale. Also, the scale’s reliability is sufficient but low.
Results: It is not clear why/how you combined believability in news among real and fake news in your first analyses. It seems that throughout your tests, your results need to be separated by belief in fake news and belief in real news. There is big difference in believing real news and believing fake news.
Discussion: Instead of mostly reiterating the results, as you state in your abstract that the study holds insights “for researchers, practitioners, social media users, and social media platform providers” explicate what those insights are. You should also consider deeper insights and solutions for news consumers. In your limitation section, given that your sample is heavily female, also address gender differences in empathy. Can you also report overall gender differences? Did males or females have higher empathy and believability in fake news?
Comments on the Quality of English LanguageAs I believe this manuscript was translated from Chinese to English, it needs a close read for misuse of words and clarity.
Author Response
Responses to Reviewer 1:
Comments and Suggestions for Authors:
Thank you for the opportunity to review your manuscript examining the role of empathy in susceptibility to believing fake news. As interest in the research area of emotion and online news processing continues to grow, especially in regards to acceptance of misinformation, I believe there is value to your study. However, the writing and framing of the rationale for your study and the evidence used in the manuscript both need to be strengthened. Simply relying on the argument that empathy has not widely been studied in the context of believing fake news is not enough. Furthermore, there is confusion as to whether the goal of your study is to explain the internal process of empathy or to examine the impact of empathy in believing false news. Your manuscript focuses disproportionality on modeling an internal process of empathy that is already known. If indeed, you are interested in the connection between emotion and news belief, you need to more fully address the areas of literature given below in the introduction of your manuscript and test and report differences in real and fake news believability throughout your results.
Response: Thank you very much. We have revised the manuscript according to your specific comments and hope the revised draft has responded well to your comments.
Specific Comments:
Introduction/Literature Review:
Introduction 1:
Emotion and News Research: In the area of emotion and news, researchers have been interested in 1) how the emotional framing of news content influences readers. For example, research confirms that audiences are more likely to believe emotional news, even if it is false. 2) The effects of experiencing specific emotions on belief in fake news. Your study falls into this latter category. You need to make this clear. At the very end of your manuscript, you say that this about “being exposed to high-state empathetic news” (line 384). You blur the emotion of the news consumer and the emotion of the news here.
Response:
We appreciate such detailed comments and good suggestions that may help us improve this work. According to your suggestions, the revised manuscript has made the following changes: 1) added “the effects of experiencing specific emotions on belief in fake news, i.e.,” (lines 50-51) in the fourth paragraph of the introduction to clarify the focus of this study. 2)“especially when exposed to high-state empathetic news.” (line 391) should be corrected to “especially when exposed to news that triggers high state empathy.” (lines 414-415)
Introduction 2:
Emotion and Belief in Fake News: First, an extensive body of emotion literature explains the differential impact of negative and positive mood states and specific discrete emotions on cognition and decision-making. For example, literature on the relationship between emotion and gullibility has found that a negative mood state generally increases skepticism, whereas a positive mood state increases gullibility and decreases the ability to detect deception (Forgas and East 2008; Forgas 2019). Then, make it clear that specifically in the area of fake news belief, several studies have suggested that people who engage in more reasoning are less likely to fall for fake news. Martel et al. (2020) found heightened emotionality in participants was predictive of greater belief in fake (but not real) news posts. Preston et al. (2021) found people with higher emotional intelligence, who were more emotionally perceptive, were less likely to fall for fake news. “This finding supports the idea that high-EQ individuals are more likely to be able to see beyond the emotionally charged content which is a hallmark of fake news, leading to a more effective critical evaluation of the likely veracity of the content.” [https://www.ncbi.nlm.nih.gov/pmc/articles/PMC7951906/]. The conclusion here is that higher order thinking and emotional detection lead to less belief in fake news.
Response:
We thank the reviewer for pointing out this important detail, which we overlooked. We fully agree with the reviewers’ suggestion that We should supplement the influence of emotion on cognition and decision-making, increase the discussion of emotion and susceptibility, and make the logic more rigorous.
1) The specific content added to the introduction is as follows:
“An extensive body of literature on emotion explains the differential impact of negative and positive mood states and specific discrete emotions on cognition and decision-making. Literature on the relationship between emotion and belief found that emotions induced by news reading can influence belief [5,16]. For instance, a negative mood state generally increases skepticism, whereas a positive mood state increases gullibility and decreases the ability to detect deception [65,66]. Martel et al. [11] found that heightened emotionality in participants predicted greater belief in fake (but not real) news posts. (See pages 3, lines 97-104)
Reference:
- Forgas, Joseph P., and Rebekah East. "On being happy and gullible: Mood effects on skepticism and the detection of deception." Journal of Experimental Social Psychology44, no. 5 (2008): 1362-1367.
- Forgas, Joseph P. "Happy believers and sad skeptics? Affective influences on gullibility." Current Directions in Psychological Science28, no. 3 (2019): 306-313.
2) The following content has been added to Discussion:
“It seems to be inconsistent with the results of Preston et al. [12]. Preston et al. (2021) found people with higher emotional intelligence, who were more emotionally perceptive, were less likely to fall for fake news. “This finding supports the idea that high-EQ individuals are more likely to be able to see beyond the emotionally charged content which is a hallmark of fake news, leading to a more effective critical evaluation of the likely veracity of the content.” The conclusion is that higher-order thinking and emotional detection lead to less belief in fake news. This inconsistency may be related to the differences in the specific methods of the two studies. Our focus is trait empathy, which is not the same as emotional intelligence. And, as mentioned in the Introduction, their results may be mixed with the effects of analytical thinking. Hoffman [67] views empathy as a largely involuntary vicarious response to affective cues from another person or her situation, our findings, that the irrational component of trait empathy is related to online news belief rather than its rational component, also suggest the possible influence of analytical thinking. Therefore, the results of the two studies may not be completely contradictory.” (Please see page 9, lines 310-323)
Reference:
- Hoffman, Martin L. "Is altruism part of human nature?." Journal of Personality and social Psychology40, no. 1 (1981): 121.
Introduction 3:
The Role of Empathy in News Processing: You need to better explain the automatic and more cognitive aspects (perspective taking) of empathy [see: https://pubmed.ncbi.nlm.nih.gov/15537986/] and how the appraisal pattern of this positive, prosocial, “every day mind reading” emotion led to your expected predictions and results. It seems there is reason to believe that the approach, positive mindset of empathy will lead to more susceptibility to fake news, especially for the more automatic affective empathy process and in-the-moment state empathy, as you suggest. Currently, your use of emotional contagion theory is not sufficient. You need to go deeper into the specific emotion of empathy. This is likely not just about being more “emotionally responsive” as you state on line 324.
Response:
Thank you for the constructive comments and suggestions. We first added an explanation of the relationship between empathy and news processing, and then we further added to the discussion how various aspects of empathy led to our results.
The specific content added to the introduction is as follows:
1) we added “Similar to many emotion-related processes, some empathy components occur implicitly and sometimes without awareness. State empathy in news reading involves such emotional processing. News reading may cause the phenomenon of emotional contagion, the tendency to automatically mimic and synchronize facial expressions, vocalizations, postures, and movements with those of another person and, consequently, emotional convergence with the other [64].” (See pages 2 - 3, lines 93-98)
2)we deleted “ Research examining emotions has shown that emotional traits generally correlate positively with emotional states and have the propensity to experience related emotional states [15]. It was also demonstrated that emotions induced by news reading can influence belief [5,16]. For instance,”
Reference:
[64] Hatfield, E., Cacioppo, J. T., & Rapson, R. L. (1993). Emotional Contagion. Current Directions in Psychological Science, 2(3), 96-100.
Introduction 4:
Hypotheses: Make is clear whether you predict the influence of empathy will relate to an increase or decrease in susceptibility to fake news specifically.
Response: Thank you for helping us improve the manuscript. We realized that the original text was somewhat vague. We have added related information according to the suggestions as follows:
In introduction 1.1, we proposed Hypothesis 1: higher scores in AE would be associated with higher belief in online news. Then, in Introduction 1.2 We proposed Hypothesis 2: SE would partially mediate the effects of AE on the belief in online news. specifically, higher scores in AE would be associated with higher SE in reading online news, and higher scores in SE would increase believability in online news.
Finally, in introduction 1.3, We proposed Hypothesis 3: The influence of SE on the belief in online news would be moderated by news type. specifically, higher scores in SE would increase believability in both fake news and real news, but the effect on fake news is even greater.
The specific content is as follows:
1) added, “ specifically, higher scores in AE would be associated with higher SE in reading online news, and higher scores in SE would increase the believability in online news.”(Page 3, lines 123-125) 2) added, “specifically, higher scores in SE would increase the believability in both fake news and real news, but the effect on fake news is even greater.”(Page 4, line 145-146)
Method 1:
Stimulus Materials: Can you provide an example of the news stimuli used? You also need to explain if the stimuli were selected to be emotionally neutral or generate emotional responses in participants and the implications of this. You state that you measured emotion response but it is not included in your descriptive results.
Response: Thank you for pointing out this important detail, which we overlooked.
Firstly, we address the question you raised. State empathy is characterized by emotional response following headline reading in our study. The stimuli were selected to generate different emotional responses in participants, we aimed to measure whether participants' emotional response (SE) to the news correlated with their belief in the news. We measured participants' state empathy for news-specific types (1 - Can't feel it at all; 7 -- Can totally feel it), Examples of the news stimuli used are as follows (Left-Fake; Right-Real). It should be noted that the stimulus materials and questionnaires are presented in Chinese:
The specific content is as follows (See page 5, lines 176-179):
We added (Figure 1 shows examples of news stimuli).
Figure 1. The News Stimuli (Left-Fake; Right-Real)
Method 2:
Measures: Provide examples items for the main IRC-C trait empathy scale. Also, the scale’s reliability is sufficient but low.
Response:
Thank you for your professional and careful advice. We have added the example items for the main IRI-C trait empathy scale. “The example items for the main IRI-C scale were as follows. PT: I try to see each person’s side of the argument before making a decision; FS: After watching a play or movie, I feel as if I were one of the characters in the play; EC: Other people’s misfortunes do not usually bother me much; PD: When I see that someone has had an accident and needs help badly, I become so nervous that I almost have a nervous breakdown.” (See page 5, lines 184-189)
In addition, as you pointed out, “the scale’s reliability is sufficient but low”, We have revised the manuscript in section 4.4, “Finally, the reliability of IRI-C, although sufficient, was relatively low.” (See page 10, lines 384-385)
Results:
It is not clear why/how you combined believability in news among real and fake news in your first analyses. It seems that throughout your tests, your results need to be separated by belief in fake news and belief in real news. There is big difference in believing real news and believing fake news.
Response:
Thank you for this important reminder. We fully agree with your point that there is a big difference between believing real news and believing fake news. But what we wanted to explore in the first analysis was whether trait empathy and its subdimensions are universal predictions of belief in news (whether real or fake). In the final 3.3 result analysis, we further distinguish the process of believing real news and believing fake news. It reveals the difference of trait empathy in the processing of real news and fake news.
Discussion:
Instead of mostly reiterating the results, as you state in your abstract that the study holds insights “for researchers, practitioners, social media users, and social media platform providers” explicate what those insights are. You should also consider deeper insights and solutions for news consumers. In your limitation section, given that your sample is heavily female, also address gender differences in empathy. Can you also report overall gender differences? Did males or females have higher empathy and believability in fake news?
Response:
Thanks for your constructive comments and suggestions.
First of all, we agree with your viewpoint that our Discussion section lacks further explanation and discussion of the results, as well as deeper insights and solutions for news consumers. Here, we have added the following paragraph to the Discussion:
“In everyday life, for empathetic individuals, when they see news about an event that is described in a lot of emotional words, they would better have the following conversation with themselves: The author seems to be describing the incident with emotion, and the truth of the incident may not be exactly like this? Is there something wrong with the content of this news? And is it possible that this is happening in reality? Such self-questioning may trigger one’s analytical thinking and avoid the increased belief in fake news caused by empathy.” (Please see page 10, lines 370-376)
Secondly, thank you for pointing out this important detail. As you emphasized, we found some gender differences. Women have higher levels of trait empathy, affective empathy, and belief in fake news. Specifically as follows: By gender grouping, the independent sample T-test examined the differences in trait empathy (TE), affective empathy (AE), cognitive empathy (CE), state empathy (SE) of fake news, real news, and believability of fake news and real news. Our results found that the trait empathy (57.27 ± 9.47 vs. 53.81 ± 9.40, t(138) = 2.046, p = 0.043), affective empathy (28.31 ± 5.74 vs. 24.70 ± 6.17, t(138) = 3.425, p < 0.001) and belief in fake news (4.71 ± 0.82 vs. 4.32 ± 0.84, t(138) = 2.61, p = 0.010) of females were significantly higher than males. The pairwise differences in the remaining variables were not significant.
The statistics of each variable are as follows:
Gender |
TE |
AE |
CE |
Type |
SE |
Believability |
Male M(SD) |
53.81(9.40) |
24.70(6.17) |
28.68(5.72) |
Fake |
4.42(0.91) |
4.32(0.84) |
|
Real |
4.86(0.86) |
5.57(0.66) |
|||
Female M(SD) |
57.27(9.47) |
28.31(5.74) |
29.41(6.52) |
Fake |
4.54(0.83) |
4.71(0.82) |
|
Real |
4.92(0.86) |
5.67(0.68) |
The interesting finding is that women have higher believability in fake news than men. According to the results, we tentatively estimate that this difference is more likely due to the role of affective empathy rather than emotional responses (SE). This is consistent with some research showing that a differentiation based on biological sex is related to susceptibility to fake news (Paula Herrero-Diz et al. 2019; Wright et al., 2022).
Reference:
Herrero-Diz, Paula, Jesús Conde-Jiménez, Alejandro Tapia-Frade, and David Varona-Aramburu. "The credibility of online news: an evaluation of the information by university students/La credibilidad de las noticias en Internet: una evaluación de la información por estudiantes universitarios." Culture and Education 31, no. 2 (2019): 407-435.
Wright, Chrysalis L., Kwame Gatlin, and Renee Rivera. "College Student's Distrust in Hard News and Exposure to Fake News During the COVID-19 Pandemic." Journal of Media Research 15, no. 1 (2022).
Comments on the Quality of English Language:
As I believe this manuscript was translated from Chinese to English, it needs a close read for misuse of words and clarity.
Response: We are grateful for your comment. We have polished the language in our revised manuscript through the editing service (MDPI Editing Services) to improve the manuscript's readability.
Reviewer 2 Report
Comments and Suggestions for Authors
I really liked the topic of this paper, as well as the way the research was developed and presented.
Obviously this work has limitations, which are very well scrutinised by the authors.
I only have one suggestion for improvement: in the keywords, instead of "state empathy; trait empathy; affective empathy; online news; belief", I would just put affective empathy; online news; fake news; news belief.
Comments on the Quality of English Language
A careful reading to eliminate some typos and punctuation flaws.
Author Response
We are grateful for your comment. We modify the keywords to “affective empathy; online news; fake news; news belief” (line 20)
Reviewer 3 Report
Comments and Suggestions for Authors
In my opinion, it is an interesting, well-described, statistically rigorous study that sheds light on how different facets of empathy affect perception of online news.
These results are extremely important in the age of fake news. At the end of the manuscript, the authors carefully discuss potential implications of their results as well as venues for future studies.
Please describe how the variable CE participated in your statistical tests. At the moment, this is bit unclear
My only recommendation is to double-check the manuscript to avoid minor grammar errors.
Comments on the Quality of English LanguageMy only recommendation is to double-check the manuscript to avoid minor grammar errors.
Author Response
Responses to Reviewer 3:
Comments and Suggestions for Authors:
In my opinion, it is an interesting, well-described, statistically rigorous study that sheds light on how different facets of empathy affect perception of online news.
These results are extremely important in the age of fake news. At the end of the manuscript, the authors carefully discuss potential implications of their results as well as venues for future studies.
Please describe how the variable CE participated in your statistical tests. At the moment, this is bit unclear
My only recommendation is to double-check the manuscript to avoid minor grammar errors.
Response:
We thank you for this important reminder. In the descriptive statistics section of 3.1, we analyzed the correlation between affective empathy (AE) and cognitive empathy (CE) and belief in news, respectively. The results found that only affective empathy (AE) was associated with belief in news. Therefore, our subsequent analysis did not continue to include cognitive empathy (CE).
Comments on the Quality of English Language:
My only recommendation is to double-check the manuscript to avoid minor grammar errors.
Response:
We are grateful for your comment. We have polished the language in our revised manuscript through the editing service (MDPI Editing Services) to improve the manuscript's readability.